



# Validation of Aeolus wind products above the Atlantic Ocean

Holger Baars[1], Alina Herzog[1,2,3], Birgit Heese[1], Kevin Ohneiser[1,2], Karsten Hanbuch[1], Julian Hofer[1], Zhenping Yin[1,4,5], Ronny Engelmann[1], and Ulla Wandinger[1]

[1]Leibniz Institute for Tropospheric Research (TROPOS), Leipzig, Germany
[2]Institute for Meteorology, Universität Leipzig, Leipzig, Germany
[3]now at: LEM-Software, Ingenieurbüro für Last- und Energiemanagement, Leipzig, Germany
[4]School of Electronic Information, Wuhan University, Wuhan, China
[5]Key Laboratory of Geospace Environment and Geodesy, Ministry of Education, Wuhan, China

**Correspondence:** Holger Baars (baars@tropos.de)

**Abstract.** In August 2018, the first Doppler wind lidar in space called ALADIN was launched on-board the satellite Aeolus by the European Space Agency ESA. Aeolus measures horizontal wind profiles in the troposphere and lower stratosphere on a global basis. Furthermore, profiles of aerosol and cloud properties can be retrieved via the high-spectral-resolution lidar (HSRL) technique. The Aeolus mission is supposed to improve the quality of weather forecasts and the understanding of
atmospheric processes.

We used the chance of opportunity to perform a unique validation of the wind products of Aeolus by utilizing the *RV Polarstern* cruise PS116 from Bremerhaven to Cape Town in November/December 2018. Due to concerted course modifications, six direct intersections with the Aeolus ground track could be achieved on the Atlantic Ocean, west of the African continent. For the validation of the Aeolus wind products, we launched additional radiosondes and used the EARLINET/ACTRIS lidar
Polly[XT] for atmospheric scene analysis. The six analyzed cases proof the concept of Aeolus to be able to measure horizontal wind speeds in the nearly West-East direction. Good agreements with the radiosonde observation could be achieved for both Aeolus wind products - the winds observed in clean atmospheric regions called Rayleigh winds and the winds obtained in cloud layers called Mie winds according to the responsible scattering regime. Systematic and statistical errors of the Rayleigh winds were less than 1.5 m/s and 3.3 m/s, respectively, when comparing to radiosonde values averaged to the Aeolus vertical
resolution. For the Mie winds, a systematic and random error of about 1 m/s was obtained from the six comparisons in different climate zones. However, it is also shown that the coarse vertical resolution of 2 km in the upper troposphere which was set in this early mission phase two months after launch led to an underestimation of the maximum wind speed in the jet stream regions. Summarizing, promising first results of the first wind lidar space mission are shown and proof the concept of Aeolus for global wind observations.

# 1 Introduction

On 22 August 2018, the European Space Agency (ESA) launched the Earth Explorer Mission Aeolus. This mission aims to demonstrate significant improvements in weather forecasting by measuring height-resolved wind profiles in the troposphere and lower stratosphere in order to advance the understanding of atmospheric dynamics and climate processes (ESA, 2019a). The



satellite Aeolus belongs to the ESA Earth Explorer Core Missions and has one instrument on-board, namely the Atmospheric

Laser Doppler Instrument (ALADIN). ALADIN is the first lidar (LIght Detection And Ranging) instrument on a European satellite. It is also the first space-borne instrument capable of measuring vertical profiles of wind on a global basis. Next to wind measurements, aerosol properties can be obtained as a spin-off product (Ansmann et al., 2007; Flamant et al., 2008) via the High Spectral Resolution Lidar (HSRL) technique (Wandinger, 1998; Eloranta, 2005), which is a space-borne novelty as well. Thus, one of the mission goals is to proof the concept of the new technology in space.

For precise weather forecast, the numerical weather prediction (NWP) models rely on the data assimilation of worldwide meteorological observations. But the global meteorological observing system does not provide equally distributed wind observations in time and space. The global and vertical direct wind observations that are assimilated at the European Centre for Medium-Range Weather Forecasts (ECMWF) in late 2016 (ECMWF, 2018) are mainly made by aircrafts, radiosondes, and Atmospheric Motion Vectors (AMV). AMV describe the method of observing the movement of objects (like clouds) from

space and deriving the wind velocity from its movement. But the coverage in the lower stratosphere is poor as the AMV method only provides wind information in the cloudy troposphere for the uppermost cloud level and there are only few aircraft and radiosonde measurements in the lower stratosphere. Furthermore, the main input of aircraft measurements is obtained in Europe and the USA and is not globally distributed. The global meteorological observing system is therefore suffering from a shortage of observation in specific regions, especially in the Southern Hemisphere, in the lower stratosphere, and over the

Oceans. The aim of Aeolus is to fill these gaps by providing global horizontal wind profiles in altitudes from 0 km to 30 km, ready for data assimilation in NWP models (Horányi et al., 2015a, b).

Within the German initiative EVAA (Experimental Validation and Assimilation of Aeolus observations, e.g., Baars et al. (2019) and Geiß et al. (2019)), Calibration/Validation (CAL/VAL) activities of this space mission have been performed by the Ludwig-Maximilians-University of Munich (LMU), the Leibniz Institute of Tropospheric Research (TROPOS), the German

Meteorological Service (Deutscher Wetterdienst, DWD), and the DLR (Deutsches Zentrum f. Luft- und Raumfahrt). The aim of EVAA is to validate the wind and aerosol products of Aeolus and to quantify the benefits of these new measurements for weather forecasting by assimilation experiments.

As one part of these activities, the regular participation of TROPOS on *RV Polarstern* (Knust, 2017) cruises within the OCEANET project (Macke et al., 2010; Kanitz et al., 2013; Rittmeister et al., 2017; Bohlmann et al., 2018) offered the unique

opportunity to perform ground-based validation above the Atlantic Ocean where only few observational data is available. The *Polarstern* cruise PS116 from Bremerhaven, Germany to Cape Town, Republic of South Africa, took place from 10 November 2018 to 11 December 2018 (Hanfland and König, 2019) shortly after the launch of the satellite. Starting in the northern mid-latitudes and ending in the southern subtropical region at a latitude of -33.92°, PS116 covered the northern mid-latitude region with frequent westerly winds, the subtropical jet stream region, the trade winds region, the Inter-Tropical Convergence

Zone (ITCZ), and finally ended up in the subtropical region around Cape Town.

The wind validation could be realized using radiosonde launches provided by the German Meteorological Service DWD on *RV Polarstern* (Schmithüsen, 2019). We also utilized the multiwavelength-Raman-polarization lidar Polly$^{\mathrm{XT}}$ (Engelmann et al., 2016; Baars et al., 2016) in order to characterize the atmospheric state above *RV Polarstern* which is part of the European





Infrastructure EARLINET/ACTRIS (European Aerosol Research Lidar Network/European Research Infrastructure for the
observation of Aerosol, Clouds and Trace Gases).

## 2   Wind lidar mission Aeolus

In 1999, ESA has chosen the Atmospheric Dynamics Mission (ADM, Stoffelen et al. (2005)) as the 2[nd] Earth Explorer Core
mission. The name Aeolus was inspired by the keeper of the wind in Greek mythology (Ingmann and Straume, 2016). ALADIN,
the instrument on board, is a High Spectral Resolution (HSR) elastic backscatter lidar with a Nd-YAG laser operating at
a wavelength of around 355 nm (Andersson et al., 2008; Reitebuch, 2012; Ingmann and Straume, 2016; Lux et al., 2020;
Witschas et al., 2020). The laser pulses are circularly polarized and are emitted with a frequency of 50.5 Hz. The wind profiles
are obtained from backscattering processes of the laser light pulses at moving air molecules and particles (Stoffelen et al.,
2006; Tan et al., 2008; Reitebuch, 2012; Rennie et al., 2020). The signals are separately detected by two different receiver
channels, the Rayleigh channel for backscattering from molecules and the Mie channel for backscattering from particles.
As a consequence, two independent wind measurements can be obtained. Furthermore, it gives the possibility to subsequently
perform aerosol measurements, providing the particle extinction and the particle backscatter coefficient independently (Flamant
et al., 2017; Ansmann et al., 2007; Flamant et al., 2008; Martinet et al., 2018; Flament et al., 2019).

Aeolus has a weekly repeating, polar-sun-synchronous orbit with an inclination of 97° and a mean altitude of 320 km (Kanitz
et al., 2019a; Straume et al., 2019). One orbit period has a duration of about 90 minutes (ESA, 2018; Straume et al., 2019; von
Bismarck et al., 2019). The ground track velocity is about 7200 m/s. The line-of-sight (LOS) describes the field-of-view in
which the backscattered light from the emitted laser pulses can be collected by the lidar telescope. It has an angle of 35° versus
nadir, to be able to measure the horizontal line-of-sight (HLOS) wind velocity. It should be mentioned that this angle changes
to 37.6° on the measurement ground point due to the earth curvature (Reitebuch et al., 2014).

Besides in strong convection cases, the vertical wind velocity is small compared to the horizontal wind. Thus, the vertical
wind component is neglected for calculating the horizontal wind speed from the Aeolus LOS observations. Furthermore, the
LOS is orthogonal to the flight direction in order to minimize the effect of the satellite velocity on the wind measurements. The
orbit is aligned such that Aeolus flies along the day/night border facing towards the night side to minimize the solar background
radiation (Kanitz et al., 2019a). Thus, the overpasses are either in the morning (descending orbit) at around 6 am or in the
evening (ascending orbit) at around 6 pm local time. Passing from North to South in the morning, Aeolus' viewing direction
has an azimuth angle of around 100°. This leads mainly to a measurement of the horizontal West-East wind component,
having a positive sign for easterly winds along the HLOS. Consequently, the sign is vice versa for the Aeolus track from South
to North, having an azimuth angle of around 260°.

For the Aeolus mission, the accumulation of the return signal of 19 outgoing laser pulses is defined as one *measurement* and
corresponds to a horizontal length of around ≈ 3 km. One *observation* is the average of several *measurements* and is aimed
to be about 30 *measurements* ≈ 87 km horizontal resolution for Rayleigh wind *observations*. The number of *measurements*
included in one *observation* can be modified, depending on the desired integration length. The receiver has 24 vertical range





bins and wind profiles can be obtained between 0 km and 30 km with a vertical resolution between 250 m to 2 km (Reitebuch et al., 2014).

Aeolus, i.e., ALADIN, is able to retrieve wind retrievals from Doppler shift at particles, these are the so-called **Mie winds**, but also in clean atmosphere due to the Doppler shift at molecules, these are the so-called **Rayleigh winds**. The technique on-board the satellite and the respective algorithms to retrieve the wind are described in Stoffelen et al. (2006); Andersson et al. (2008); Tan et al. (2008); Reitebuch et al. (2009); Reitebuch (2012); Ingmann and Straume (2016); Reitebuch et al. (2014); Rennie et al. (2020).

Products of Aeolus are delivered at several data levels (Reitebuch et al., 2014; Ingmann and Straume, 2016; Flamant et al., 2017). For the end user, only Level 2 is of interest on which all necessary calibration and instrument correction have been performed. The calibrated and fully processed HLOS wind is delivered in the Level 2B data (Rennie et al., 2020; Ingmann and Straume, 2016). This is the main wind product of Aeolus. There is also Level 2C data, which is vector wind data, resulted from ECMWF model analysis after the assimilation of Level 2B profiles. In Level 2A, the aerosol and cloud spin-off products (optical properties) are delivered (Ansmann et al., 2007; Flamant et al., 2008, 2017; Flament et al., 2019), which will not be discussed in this paper.

The observational requirements (Ingmann and Straume, 2016) for the Aeolus Mission are that the vertical resolution shall achieve 500 m in the Planetary Boundary Layer (PBL), 1 km in the troposphere, and 2 km in the lower stratosphere. The requirements for the horizontal integration length per *observation* depends on the measurement type and altitude. The precision of the HLOS component is aimed to be 1 m/s within the PBL, 2.5 m/s for the troposphere, and 3 m/s for the lower stratosphere. As the Aeolus observations shall be used to improve the weather forecast by data assimilation, the data must be available within 3 hours (Reitebuch et al., 2014).

## 3   Data set and Methodology

For the validation of the Aeolus wind products, the Level 2B is the product of choice for comparison to the radiosonde measurements. These are the fully calibrated and processed HLOS winds ready for data assimilation in NWP models. The output of the product includes different classifications and quality parameters which need to be chosen correctly. The use of these parameter is described in the following:

**Atmospheric classification**

The Level 2B product provides four separated wind profiles for one atmospheric scene according to the atmospheric classification performed in the processor chain (Rennie et al., 2020). These four wind "types" are:

- Rayleigh$_{clear}$: Wind derived in atmospheric regions without any particle backscatter, thus in clear sky using the Rayleigh methodology,





– Rayleigh$_{\text{cloudy}}$: Wind derived from *measurements* with non-zero particle backscatter, thus in a cloudy or particle-loaded environment using the Rayleigh methodology,

– Mie$_{\text{clear}}$: Wind derived in atmospheric regions with zero particle backscatter using the Mie methodology. As in clear sky condition no Mie wind should be detectable, this is only possible if the classification failed to detect particle backscatter,

– Mie$_{\text{cloudy}}$: Wind derived in atmospheric regions with non-zero particle backscatter using the Mie methodology.

Each range-bin of the *measurements* (2.7 km horizontal scale) in the *observation* (87 km horizontal scale = 30 *measurements*) is analyzed individually for the atmospheric scene classification. The classification can be done by using the scattering ratio, a particle feature finding algorithm or the particle extinction coefficient as criteria (Rennie et al., 2020). The currently applied method by ESA is the use of the scattering ratio. For this, a predefined scattering ratio threshold value as a function of altitude is used. If the scattering ratio is higher than the threshold value, particle scattering is considered to be dominant. Below the threshold, molecular scattering only is assumed. The range-bins assigned to the same classification type are accumulated within the corresponding *observation*. This accumulation of the *measurements* improves the signal-to-noise-ratio and provides a large-scale wind observation which is ready for the NWP data assimilation (Rennie et al., 2020). The Rayleigh and Mie wind retrieval algorithms are then applied each to both classes within the *observation*. Thus, one *observation* comprised four different wind types for each range bin, namely Rayleigh$_{\text{cloudy}}$, Rayleigh$_{\text{clear}}$, Mie$_{\text{cloudy}}$, and Mie$_{\text{clear}}$.

To sum up, each *observation* with a horizontal length of 87 km consists of individual *measurements* with a horizontal length of about 3 km. Within the *observation*, the *measurements* are grouped into the four different classifications, namely Rayleigh$_{\text{cloudy}}$, Rayleigh$_{\text{clear}}$, Mie$_{\text{cloudy}}$ and Mie$_{\text{clear}}$. As the cloud and aerosol situation is usually not homogeneous within the 87 km, only the *measurements* which are useful for the respective classification are taken into account for the wind retrieval. If, for example, a cloud layer exists in the first 21 km of the observation, the Mie$_{\text{cloudy}}$ wind product considers only the *measurements* of these first 21 km. As this procedure is not only applied to the profiles but for each vertical range bin individually, the coordinates of the Aeolus *observations* profiles given at a certain range bin can be different. While, e.g., at 4 km altitude a cloud is observed for the first 21 km, another one is observed at 7 km altitude in the last 30 km of the 87 km horizontal path. Then the coordinates given for the Rayleigh$_{\text{cloudy}}$ and Mie$_{\text{cloudy}}$ winds at 4 km altitude are the mean coordinates of the first 21 km, while for 7 km height, the mean coordinates of the last 30 km are used.

To make it even more difficult, in principle, the Mie and Rayleigh wind *observations* can have a different horizontal resolution. In this work, however, we analyzed early-mission data obtained shortly after launch during the commissioning phase of Aeolus, and at this time the horizontal resolution for both, Rayleigh and Mie winds, was equal and about 87 km. As Mie$_{\text{cloudy}}$ winds benefit from strong backscatter at cloud particles, the horizontal resolution is meanwhile increased to 12 km due to the significantly higher signal-to-noise ratio of this "wind type". The Rayleigh horizontal resolution is, however, kept at 87 km.

It is obvious that only two out of this four wind products are useful, namely the Rayleigh$_{\text{clear}}$ and the Mie$_{\text{cloudy}}$ product. For an accurate Mie wind *measurement*, a strong particle backscatter is required, whereas the best quality of the Rayleigh *measurements* is achieved in clear sky conditions. Thus, we will use only these two observation types, Rayleigh$_{\text{clear}}$ and the Mie$_{\text{cloudy}}$, for our analysis.





**Error threshold and Validity flag**

The Level 2B product provides a HLOS error estimation for each range-bin in the *observation* profiles. We only consider wind data with errors < 8 m/s for the Rayleigh$_{clear}$ and < 5 m/s for the Mie$_{cloudy}$ winds. This error threshold results from recommendations of ESA/DISC (Reitebuch et al., 2019b; Stoffelen et al., 2019; Rennie and Isaksen, 2019; Isaksen and Rennie, 2019) to the Aeolus CAL/VAL teams.


The validation flag (de Kloe et al., 2016) considers the validity of the products. It has either the value 1 (valid) or 0 (not valid). We only use Aeolus products with a validity flag of 1.

**Hot pixel**

During the commissioning phase of Aeolus, it was noticed that pixel with an increased dark current occurred in the memory

zone of both ACCD (Accumulation Charge Coupled Device) in the detector unit of ALADIN (Reitebuch et al., 2019a; Kanitz et al., 2019b). These pixel are called hot pixel and their increase dark current can have a changing magnitude with time. As no correction procedure was available at the early mission period we focus on, we skipped all height bins at which a hot pixel occurred, as they significantly bias the Aeolus wind and aerosol products. For our analyzed data period, these are range bins 2, 13, 16, and 24 of the Mie products and range bins 5, 11, and 15 of the Rayleigh products (note that according to ESA's

nomenclature, range bin 1 is the highest and range bin 24 the lowest of the profile). It is worth noticing that meanwhile a hot pixel correction is in place for Aeolus data since 14 June 2019.

**Observation geometry**

As Aeolus provides only the wind along the HLOS, which is mainly the west-east wind component, the radiosonde measurements are projected to the HLOS of Aeolus using the following formula:

$$v_{RS_{HLOS}} = v_{RS} \cdot \cos(\varphi_{Aeolus} - \varphi_{RS}). \tag{1}$$

$v_{RS}$ describes the horizontal wind velocity and $\varphi_{RS}$ the wind direction measured with the radiosonde. $\varphi_{Aeolus}$ is the azimuth angle of Aeolus, which is obtained from the Level 2B data and differs depending on range-bin and global position.

## 4 Aeolus Validation

The ship-borne validation took place during the *RV Polarstern* cruise PS116 (10 November 2018 to 11 December 2018) from Bremerhaven, Germany to Cape Town, Republic of South Africa (Hanfland and König, 2019). Figure 1 shows the ground tracks of Aeolus obtained with the ESA ESOV tool along the track of the ship (white thick line). Each colour indicates a different weekday of the Aeolus overpass. Along the cruise, six points of intersection with the ground tracks of Aeolus within



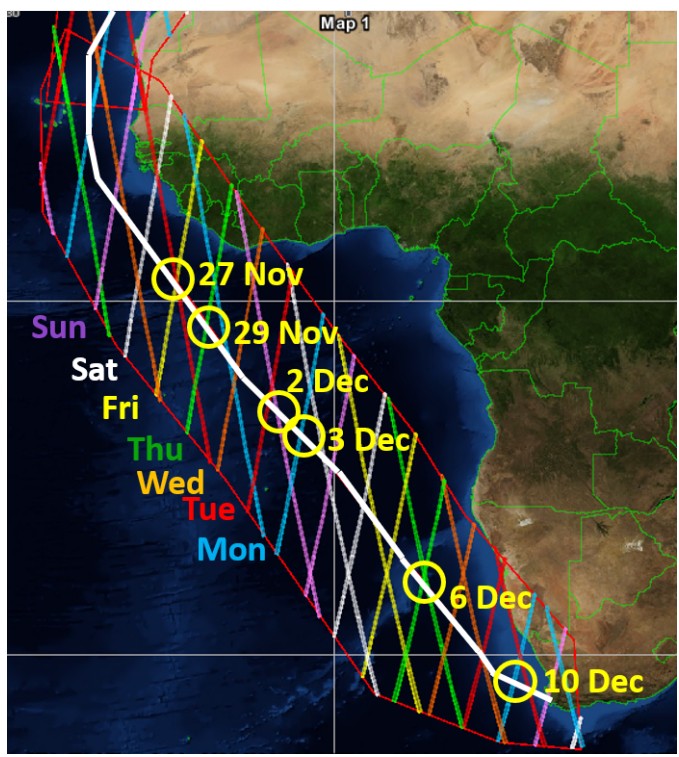

**Figure 1.** Ground tracks of Aeolus (thick coloured lines). Each colour represents another weekday as indicated in the plot. The thick white line represents the ship track of *RV Polarstern*. The yellow circles indicate the points of intersection of *RV Polarstern* and the Aeolus ground track for which additional radiosondes were launched.

a 150 km radius around *Polarstern* were possible for which additional radiosondes could be launched (yellow circles).

The radius was chosen as a compromise between the number of possible points of intersection and a reasonable limit for the distance between the two individual measurements (radiosonde vs. Aeolus profiles). For the wind validation, radiosondes of the type RS41 (Jauhiainen et al., 2014; Jensen et al., 2016) produced by the company Vaisala (Finland) and provided by the German Meteorological Service DWD (Schmithüsen, 2019) with a vertical range up to 30 km were launched one hour prior to the Aeolus overpass. An overview of the six obtained validation cases for the cruise PS116 is given in Tab. 1.

For the presented validation, we have used the Aeolus operational product (Baseline 2B02 with processor version 03.01) available at the time of the overpass. For this product, a correction of the hot pixel as mentioned above was not yet performed, thus we skip the respective height bins.





**Table 1.** Overview of Aeolus validation cases performed on-board *RV Polarstern* with radiosonde (RS) launches during cruise PS116. The date, the location of *RV Polarstern* during the RS launch, the launch time, the time of the exact Aeolus overpass, as well as the distance between the radiosonde and the closest Aeolus wind profile is given.

| Date | RS launch location | RS launch time | Aeolus overpass time | Closest distance |
|---|---|---|---|---|
| 27 Nov 2018 | 3.573° N, 14.992° W | 1759 UTC | 1901 UTC | 134 km |
| 29 Nov 2018 | 1.682° S, 10.879° W | 0528 UTC | 0634 UTC | 40 km |
| 2 Dec 2018 | 10.653° S, 3.663° W | 1729 UTC | 1831 UTC | 100 km |
| 3 Dec 2018 | 11.922° S 2.627° W | 0500 UTC | 0558 UTC | 36 km |
| 6 Dec 2018 | 22.725° S, 7.141° E | 1648 UTC | 1749 UTC | 47 km |
| 10 Dec 2018 | 31.726° S 14.156° E | 0329 UTC | 0432 UTC | 111 km |

### 4.1 Case studies for the validation of the Aeolus wind products

In the following, the performance of Aeolus will be discussed intensively by means of two dedicated case studies. The other
four validation cases are presented afterwards to provide an overview of the measurements as all are used for a statistical analysis presented in the respective section. An overview of all six validation cases is given in Tab. 1.

#### 4.1.1 Case study 1: 29 November 2018

In the first case study, the observations of 29 November 2018 (see Fig. 1) which are representative for a tropical wind regime are discussed. For this case, the Aeolus ground track could be reached within a distance of $\approx 40\,\mathrm{km}$ (Tab. 1) after *RV Polarstern*
had just passed the equator.

Also, on-board *RV Polarstern* was the portable multiwavelength-Raman-polarization lidar Polly$^{\mathrm{XT}}$ of the OCEANET facility (Engelmann et al., 2016). With its setup, aerosol and cloud properties can be classified by shape, size, and absorption behaviour (Baars et al., 2016, 2017). The observations with this EARLINET/ACTRIS lidar will be used to characterize the atmospheric state above *RV Polarstern*. Figure 2 shows the temporal evolution of the attenuated backscatter coefficient (calibrated range-
corrected signal) to get an impression of the atmospheric scenery for the time around overpass at 0630 UTC. A cloud layer at around 2 km was observed exactly during the Aeolus overpass (red rectangle). The lidar could not penetrate this optically thick cloud. Below this cloud, the marine boundary layer (BL) was located as indicated by moderate backscattering (green colours). Below 400 m, no signal was obtained due to the incomplete overlap between the receiver field of view and the laser beam of the lidar. Having a look at the period after the Aeolus overpass without cloud occurrence (after ca. 0725 UTC), an aerosol layer
up to around 4 km is visible (greenish-bluish colours).

As the radiosonde drifts along the wind direction, the distance to Aeolus changes during the measurements. This is illustrated for this case study in Fig. 3 for the two closest Rayleigh (green and blue) and Mie (purple and cyan) *observation* profiles. While the horizontal distance to the Mie$_{\mathrm{cloudy}}$ profiles varies between 10 km and 55 km in the lower 5 km (remember the accumulation of *measurements* within one *observation* as discussed above), the distance to the Rayleigh$_{\mathrm{clear}}$ profile has only minor changes

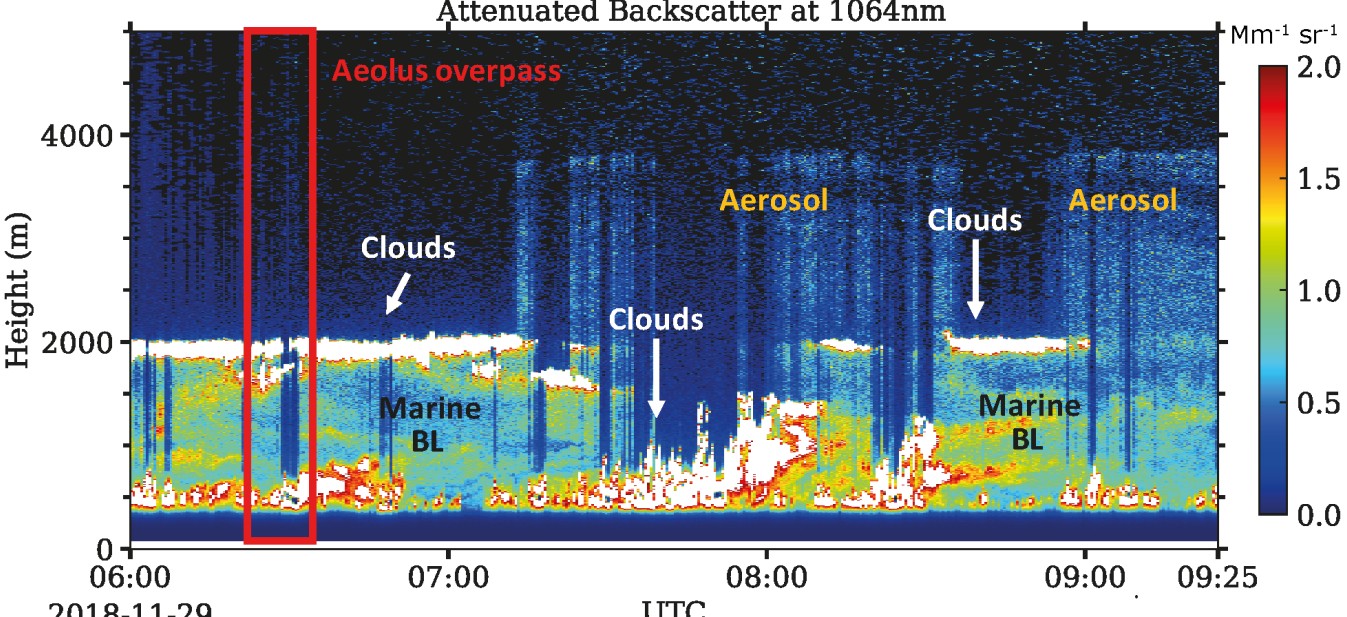

**Figure 2.** Time-height plot of the attenuated backscatter coefficient at 1064 nm around the time of the Aeolus overpass (red rectangle) on 29 November 2018.

and is on average as indicated in the legend of Fig. 3. The distance changes are not only caused by the radiosonde drift, but in particular because of the Aeolus classification algorithm as discussed above.

Figure 4 shows the HLOS wind velocity profiles measured by the radiosonde (red) launched for this overpass and the two closest Aeolus Rayleigh$_{clear}$ (green and blue) and Mie$_{cloudy}$ profiles (magenta and cyan). Figure 4a provides the radiosonde profile with its highest vertical resolution while in Fig. 4b, the vertical resolution of the radiosonde measurements is aggregated

in order to match exactly the Aeolus range-bin setting. The given distance in the legend of Fig. 4 is the mean distance regarding all range bins. The uncertainty estimation of the radiosonde wind velocity profile is based on calculations of the Global Climate Observing System Reference Upper-Air Network (GRUAN), which estimated an uncertainty between 0.4 m/s to 1 m/s for the wind velocity and 1° for the wind direction (Dirksen et al., 2014). Even though this reference considers the Vaisala radiosonde type RS92 and not RS41, which was used on *RV Polarstern*, there is no significant difference in the uncertainty, as both

radiosonde types are based on the same technique to derive wind velocity and direction (Jensen et al., 2016).

Regarding the Mie$_{cloudy}$ profiles in Fig. 4, only measurements at the altitudes of the cloud layer between 1.5 km to 2 km (see Fig. 2) were obtained. Below the cloud, ALADIN could not receive any signal as the cloud layer was optically too thick to be penetrated by the laser beam. In Fig. 4a, it can be seen that the Mie$_{cloudy}$ measurements are in very good agreement with the radiosonde measurements. Considering the horizontal distance of both observations, one can assume that the cloud observed

above *Polarstern* was horizontally homogeneous as well as the horizontal winds in the lowermost troposphere. In Fig. 4b, a deviation of the adapted low-resolution radiosonde observations to the Mie$_{cloudy}$ measurements at the altitude of 2.5 km is





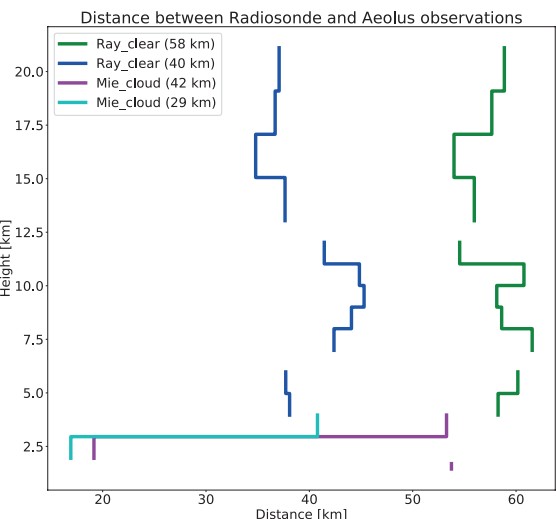

**Figure 3.** The distance between the radiosonde and the geolocation of the single wind observations for the two closest Mie$_{\text{cloudy}}$ (magenta and cyan) and Rayleigh$_{\text{clear}}$ (green and blue) profiles.

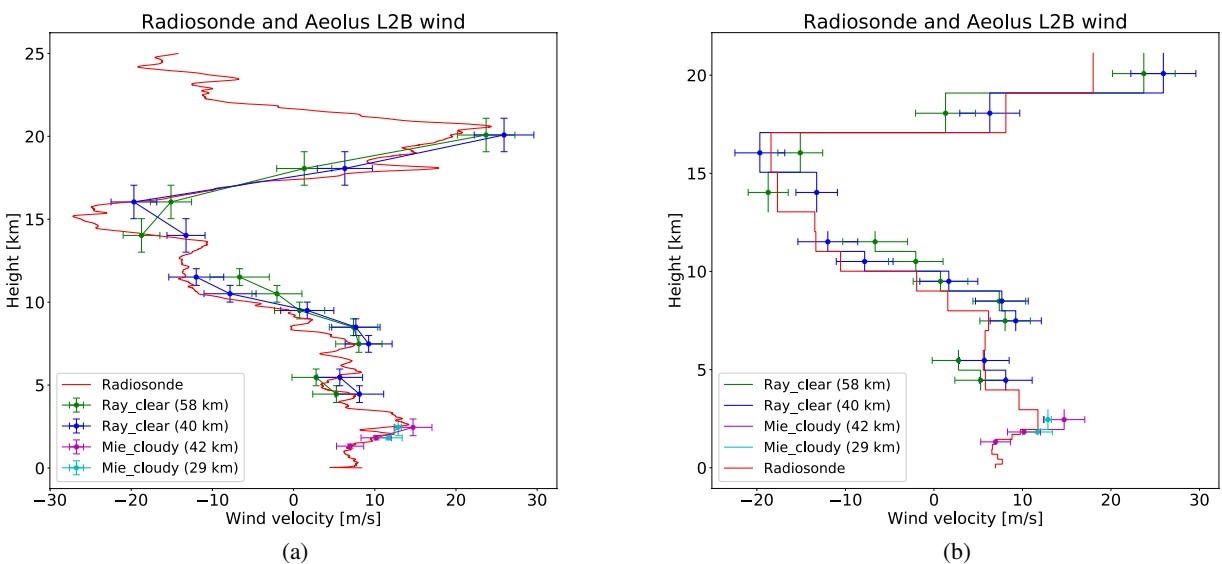

**Figure 4.** Wind velocity profiles measured by the radiosonde (red) with the two closest Aeolus Level 2B (L2B) Rayleigh$_{\text{clear}}$ (green and blue) and Mie$_{\text{cloudy}}$ profiles (magenta and cyan) on 29 November 2018. The radiosonde profile is shown with its highest resolution (a) and with an adjusted resolution to the Aeolus range-bin width (b). The radiosonde measurements are projected to the HLOS of Aeolus.



obvious. The reason for this seems to be the small-scale vertical wind variations observed by the radiosonde (short and rapid decrease of the wind velocity at around 2.5 km). In contrast, it was obviously not detected by the Aeolus measurement due to the fact that the Mie$_{\mathrm{cloudy}}$ wind is obtained practically only from return signals of the cloud and thus only from the height 235 range at which the cloud was observed within this one range bin of 1 km thickness.

Regarding the Rayleigh$_{\mathrm{clear}}$profiles, a good agreement was found for the winds between 4 km and 6 km while a positive bias (systematic error) in the region between 7.5 km to 12 km was observed for the two closest *observations*. Above 12 km, a good agreement is found, considering also the extent of the range-bins at this altitude of about 2 km. Below 4 km, no measurements are available due to the low signal-to-noise ratio and the cloud layer at 2.5 km. Summarizing, the Rayleigh$_{\mathrm{clear}}$ winds follow 240 well the shape of the wind profile from 4 km to 20 km.

Nevertheless, as can be seen in Fig. 4a, the maximum wind velocity occurs just below the tropopause at around 15 km, having an opposite direction (westerly winds) than in the lower troposphere (easterly winds). A maximum absolute wind velocity higher than 25 m/s was observed in this height region according to the high resolved radiosonde profile. However, the Rayleigh$_{\mathrm{clear}}$ wind measurements of Aeolus are not able to detect such high wind speeds. The reason for that is the low range 245 resolution of the Aeolus measurements in the higher troposphere/lower stratosphere at this time of the mission as can been seen in Fig. 4b. Here it gets obvious, that the resolution is simply too low in order to recognize the strong wind velocity in a vertically narrow atmospheric layer. In this height region, the high-resolution radiosonde wind speed (Fig. 4a) is about 8 m/s higher than compared to the radiosonde velocity aggregated to the range-bin setting of Aeolus (Fig. 4b).

At that time of the mission, i.e. shortly after the launch, the Aeolus range-bins had a resolution of 250 m up to 2 km height 250 to perform necessary ground echo characterizations. Above this height, the vertical resolution was 1 km up to the altitude of 13 km and then set to 2 km for higher altitudes as a consequence of the limitation of 24 range bins in total. Thus, considering the vertical binning, the Aeolus observations are correct, while they miss important information on the tropical jet stream speed as impressively shown here in this one example. As a consequence, the range-bins were changed to a resolution of 1 km up to an altitude of 19 km on 26 February 2019 to provide the NWP models a much more detailed wind information in a height 255 region very important for weather forecast.

Figure 5 provides an overview of the Aeolus wind profiles along the ground track during the overpass on 29 November 2018 as visualized with ESA's VirES tool (https://aeolus.services/, Santillan et al. (2019)). The location of *Polarstern* during the radiosonde launch is indicated as yellow pin. On the left side, the Rayleigh$_{\mathrm{clear}}$ wind *observations* are shown, while on the right side, the Mie$_{\mathrm{cloudy}}$ *observations* are plotted.

Noticeable is the good coverage of the Rayleigh$_{\mathrm{clear}}$ winds above 3–4 km above sea level (a.s.l.) along the whole track. The pronounced tropical jet with westerly winds as observed by the radiosonde is seen in all Rayleigh$_{\mathrm{clear}}$ *observations* as prominent feature (reddish colours). In the lower troposphere, easterly winds prevail (bluish colours) throughout the whole region. Mie$_{\mathrm{cloudy}}$ winds are available only in the lowermost 3 km where low-level clouds occurred and sporadically at high altitudes most probably due to the occurrence of cirrus clouds. The Mie$_{\mathrm{cloudy}}$ winds show steady easterly winds at the cloud 265 layer at around 2.5 km in agreement with the Rayleigh$_{\mathrm{clear}}$ winds as discussed above. However, a short statement is needed for the obviously strong westerly winds just above these easterly winds at an altitude of 3 km. These westerly winds are simply an

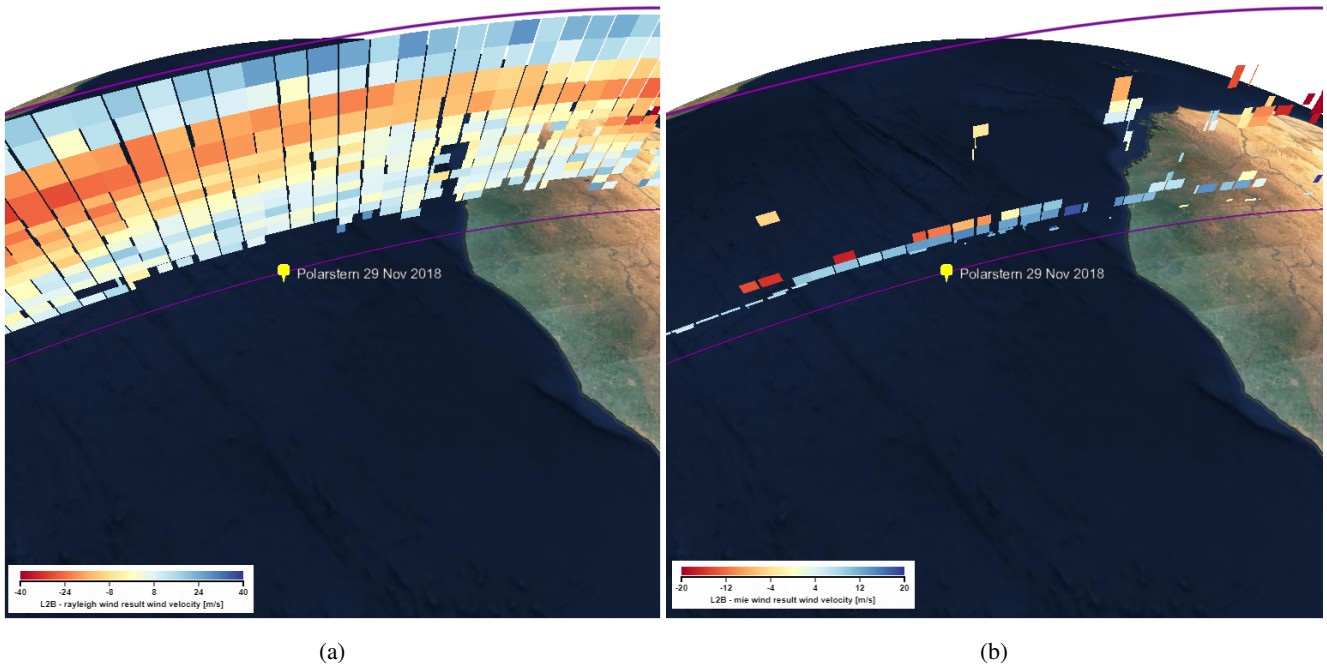

(a)            (b)

**Figure 5.** Aeolus HLOS winds on 29 November 2018. Rayleigh$_{clear}$ (a) and Mie$_{cloudy}$ (b) HLOS winds are shown for the overpass around 0634 UTC. The location of *RV Polarstern* 800 km off the shore of Liberia is indicated by the yellow pin. The figure was created with the Aeolus visualization tool VirES (https://aeolus.services/).

artefact caused by the hot pixel at range bin 13 which was left out in the analysis presented in Fig. 4 but is visualized by VirES in Fig. 5. Thus, these wind measurements at this altitude should be neglected until the hot pixel correction is in place.

### 4.1.2   Case study 2: 6 December 2018

The second case study discussed in this paper is from 6 December 2018 when *RV Polarstern* was west of Namibia (see Fig. 1) and thus already in the subtropical region. The radiosonde was launched around 50 km away from the Aeolus ground track.

     The lidar observations shown in Fig. 6 indicate no clouds at all but aerosol up to 800 m around the overpass at about 1750 UTC. Low clouds with a bottom height at around 750 m a.s.l. were observed before 1500 UTC and after 2030 UTC.

     These clouds might be the reason for the two obtained Mie$_{cloudy}$ observations below 1 km a.s.l. as presented in Fig. 7. As
described above, if during the 87 km horizontal accumulation distance some *measurements* are classified as cloudy, a valid Mie$_{cloudy}$ wind is obtained for the whole *observation*. Thus, considering the distance of *RV Polarstern* to the Aeolus ground track and the Aeolus horizontal resolution together with the cloud occurrence before and after the overpass as detected with the lidar, it is quite obvious that clouds were partly existent in the Aeolus observational domain and could be used for the Mie wind retrieval.





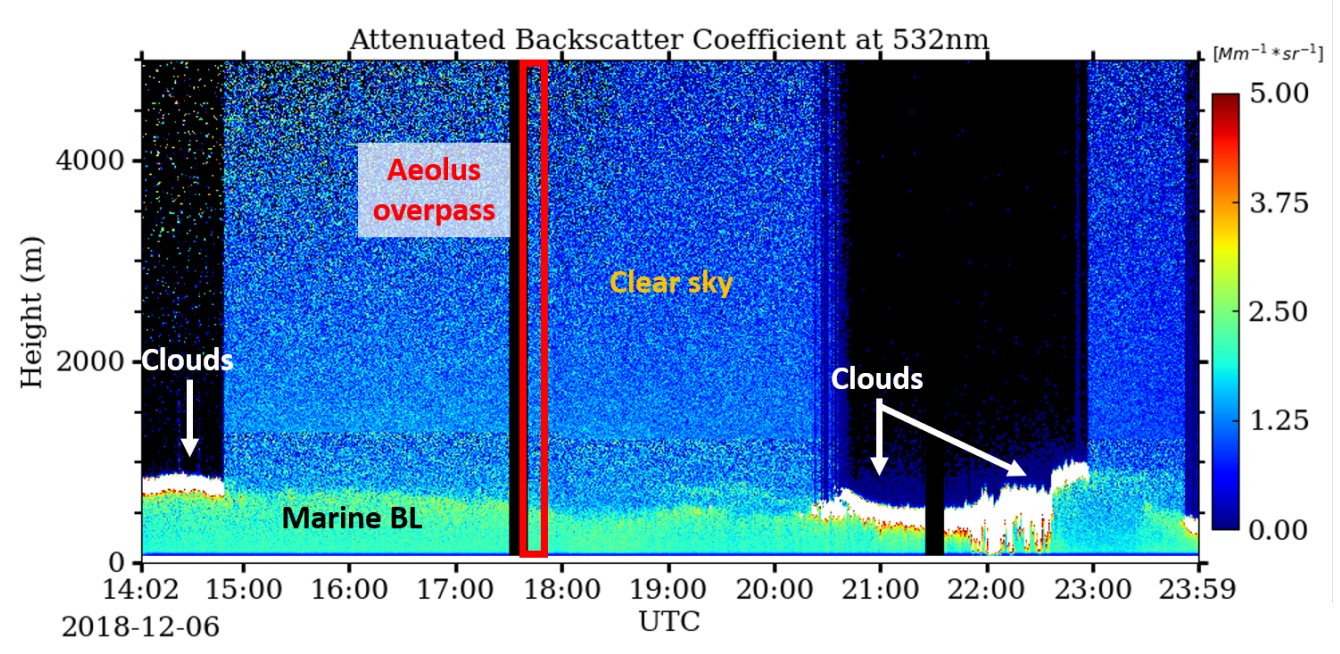

**Figure 6.** Time-height plot of the attenuated backscatter coefficient at 532 nm around the time of the Aeolus overpass (red rectangle) on 06 December 2018.

The winds observed with the Mie methodology in these as cloudy classified atmospheric regions agree perfectly with the HLOS wind obtained from the radio sounding. Also, the vertically aggregated radiosonde velocities as shown in Fig. 7b do match to the $\text{Mie}_{\text{cloudy}}$ winds due to the relatively high vertical resolution of Aeolus in the lowermost 2 km of the atmosphere.

Valid wind observations retrieved with the Rayleigh methodology are available for altitude ranges between 4 km and 21 km having its maximum at an altitude of around 10.5 km. As *Polarstern* crossed the Aeolus ground track in the evening, the

positive wind speed values in the Fig. 7 indicate westerly winds. Again, the issue concerning the low resolution of Aeolus at higher altitudes is obvious in this comparison. Even though the low resolved radiosonde measurements fit with the Aeolus ones, the high resolved radiosonde profile (Fig. 7a) shows much more and stronger changes in wind velocity, e.g., at 17 km height, compared to the low resolution one (Fig. 7b).

It is interesting to note that in case of the $\text{Rayleigh}_{\text{clear}}$ *observations*, the profile with further distance (blue line) to *RV*

*Polarstern* is in better agreement with the radiosonde measurements than the closer one (green line). Especially between 7 km and 12 km it is very similar to the radiosonde profile. Figure 8 shows the wind profiles along the Aeolus track close to *RV Polarstern*. There was a region with strong winds in higher altitudes just south of the research vessel – namely the subtropical jet. Obviously, there was a significant horizontal (north-south) gradient in high-altitude winds at the time of the overpass as seen in Fig. 8. Nevertheless, the profile represented by the green line in Fig. 7 was measured more southward along the Aeolus

track than the "blue profile". As the radiosonde drifted about 20 km to the north during its ascent, it is a logical consequence that



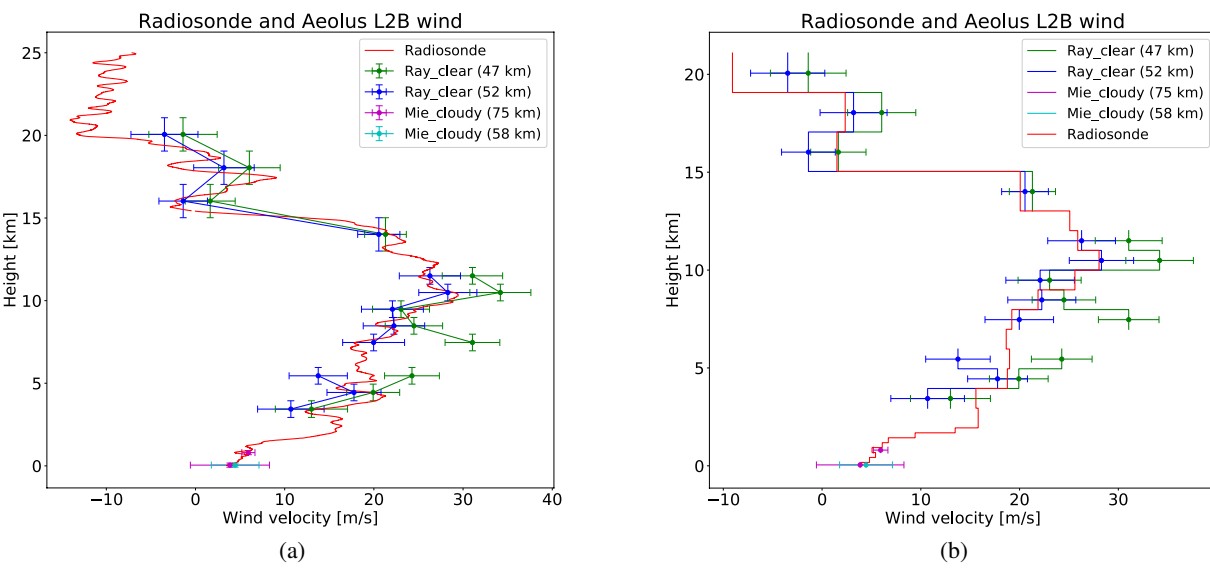

(a)                                              (b)

**Figure 7.** Wind velocity profiles measured by the radiosonde (red) with the two closest Aeolus L2B Rayleigh$_{clear}$ (green and blue) and Mie$_{cloudy}$ profiles (magenta and cyan) on 6 December 2018. The radiosonde profile is shown with its highest resolution (a) and with an adjusted resolution to the Aeolus range-bin width (b). The radiosonde measurements are projected to the HLOS of Aeolus.

the Aeolus profile measured more northerly (blue) fits better to the radiosonde. Therefore, this analysis confirms that Aeolus is well able to capture horizontal wind gradients at several heights with its Rayleigh and also Mie technique (see Fig. 8b).

### 4.1.3 Case studies 3–6

In order to provide a final overview of the validation cases obtained during the cruise, four remaining overpass cases are
presented in Fig. 9. The remaining cases are less favourable than the already presented ones due to larger distances in time and space between the research vessel location and the Aeolus observations. But they are still very valuable for the Aeolus validation in an area where almost no ground-truth observations exist. In addition, they are considered for a statistical analysis presented below.

On 27 November 2018 (Fig. 9a), the overpass region was exactly inside the ITCZ, where enhanced vertical turbulence can
occur. These vertical velocities are so far neglected in the retrieval of the Aeolus HLOS as explained above and thus might lead to higher errors in the retrieved wind speed. Furthermore, the Aeolus ground track was relatively far away from the position of *RV Polarstern* (134 km and 149 km distance at an altitude of 10 km). Valid Mie$_{cloudy}$ measurements were observed in altitudes higher than 8 km due to the existence of high clouds. Considering the large horizontal distance between the radiosonde and Aeolus profiles as well as the strong convection within the ITCZ, a reasonable agreement is found even though parts of the
Mie$_{cloudy}$ observations deviate significantly from the radiosonde observations (at around 9 km a.s.l.). The Rayleigh$_{clear}$ winds agree in shape with the radiosonde observation but the Aeolus observation at 14 km differs significantly from the radiosonde.

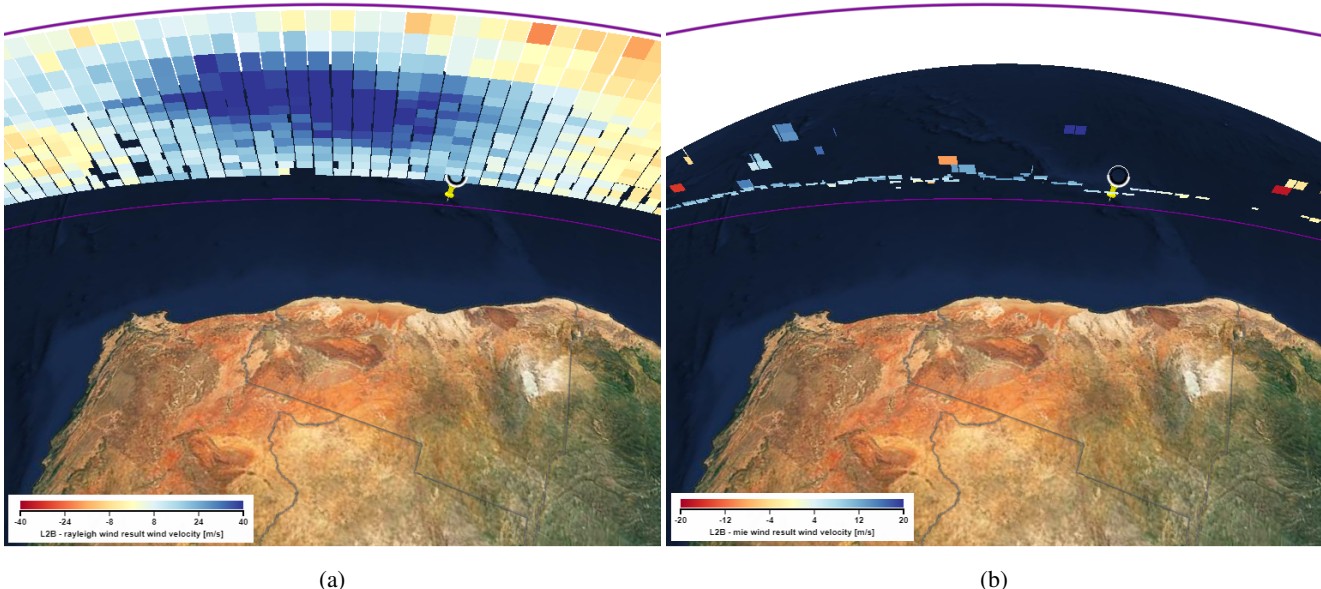

| (a) | (b) |

**Figure 8.** Aeolus HLOS winds on 6 December 2018 off the shore of Namibia. Rayleigh_clear (a) and Mie_cloudy (b) HLOS winds are shown for the overpass around 1750 UTC. Location of *RV Polarstern* is indicated by the yellow pin. The figure was created with the Aeolus visualization tool VirES (https://aeolus.services/).

From the available information, it is not possible to conclude if this strong wind speed change within a horizontal distance of 150 km is an atmospheric feature or if there are issues in the Aeolus wind retrievals. For these reasons, we excluded this case from the statistical analysis presented below.

On 2 December 2018 (Fig. 9b), the mean distance between the radiosonde and the Aeolus observations was 100 km to 122 km. The radiosonde profile shows a stronger vertical fluctuation of the horizontal wind velocity and direction than in the previously discussed case studies. Especially between 12 km to 16 km, large and fast changes of wind direction and thus the projected HLOS were observed by the radiosonde. Due to its large vertical resolution, Aeolus is only partly able to detect these rapid changes. Nevertheless, considering the vertical heterogeneity in the wind field, the agreement is acceptable for most

Rayleigh_clear wind *observations*. Aeolus-derived winds follow mostly the shape and magnitude of the radiosonde winds except for a large deviation at around 4 km (green profile). The reason for that is unclear. Probably the atmospheric classification of Aeolus was not working properly for this scene and thus cross-talk of cloud signals could have led to the deviation in the derived Rayleigh_clear winds. The observed Mie_cloudy winds, however, agree all well on this day. Mie_cloudy winds were observed at around 1 km where partly stratiform clouds were present according to the lidar measurements (not shown). Mie_cloudy winds

could also be retrieved very close to the surface and agree very well with the radiosonde observation taking into account the estimated uncertainty and the distance between the two measurements.

On 3 December 2018 (Fig. 9c), the mean distances between the closest Aeolus profiles and the radiosonde location were less than 100 km. A good agreement between the two measurements was achieved on this day. For the last point of intersection on



**Figure 9.** Wind velocity profiles measured by the radiosonde (red) with the two closest Aeolus L2B Rayleigh$_{clear}$ (green and blue) and Mie$_{cloudy}$ profiles (magenta and cyan) of all four remaining validation cases obtained during the *Polarstern* cruise - see Fig. 1.





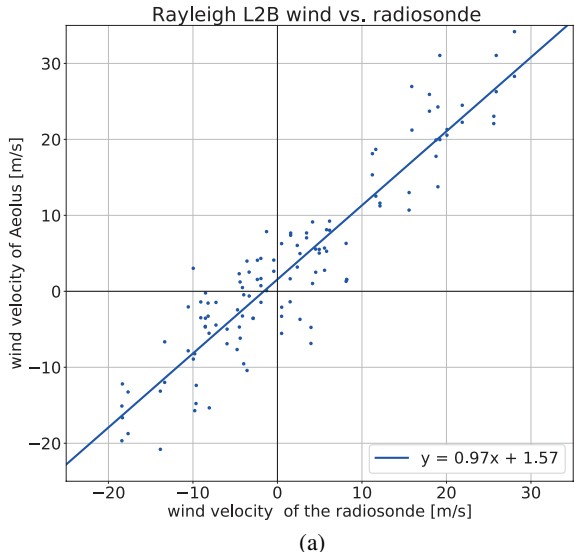
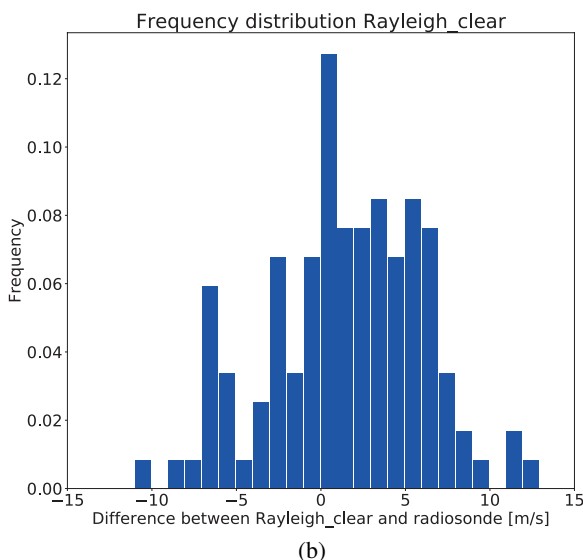

**Figure 10.** a) The L2B Rayleigh$_{clear}$ winds vs. the radiosonde measurements made during the ship-borne validation except 27 November 2018. b) Frequency distribution of the difference between Rayleigh$_{clear}$ and radiosonde wind speeds for the same data set. Radiosonde data is aggregated to the Aeolus vertical resolution and projected to the HLOS of Aeolus.

10 December 2018 (Fig. 9d), *RV Polarstern* was more than 100 km away from the Aeolus track. Like in the second case study, the Rayleigh$_{clear}$ profile which was further away is partly in better agreement with the radiosonde profile than the closer one. Also, the small-scale structures in wind speed could not be resolved by Aeolus as discussed above. Nevertheless, within the uncertainty range, a satisfying agreement was achieved for the two last case studies.

## 4.2 Statistical analysis

In this subsection, the performed comparisons are statistically analyzed. The offset between Aeolus and the radiosonde - the so-called bias - which represents the systematic error of the Aeolus wind measurements is of special interest. For this analysis, the Aeolus wind *observation* values are plotted against the corresponding values of the radiosondes averaged to the Aeolus height resolution (as discussed above) to focus on the instrumental behaviour of Aeolus only. We hereby assume that the atmospheric variability between the two measurements will not cause a bias but only increases noise, i.e., the random error. Nevertheless, the validation case of 27 November 2018 is not included in the statistics due to the large horizontal distance of the two measurements together with the fact that the observations were taken directly inside the ITCZ.

The respective correlation plot of the Rayleigh$_{clear}$ wind is shown in Fig. 10a together with the retrieved linear regression. A linear trend between the Aeolus and the radiosonde observations is clearly seen. The trend line has a slope of 0.97 with an offset (i.e., a bias) of 1.57 m/s.

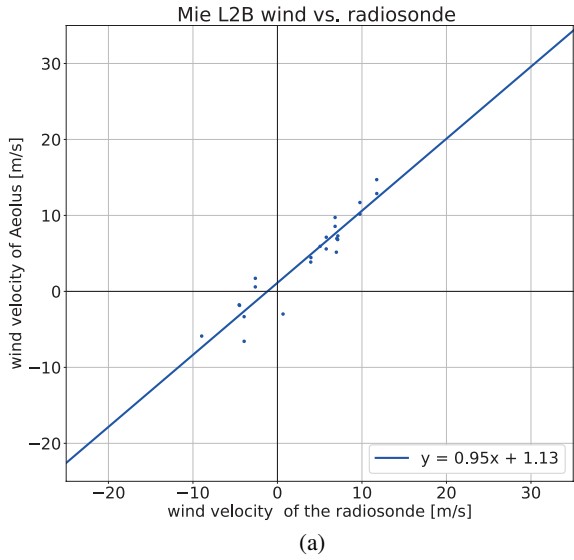
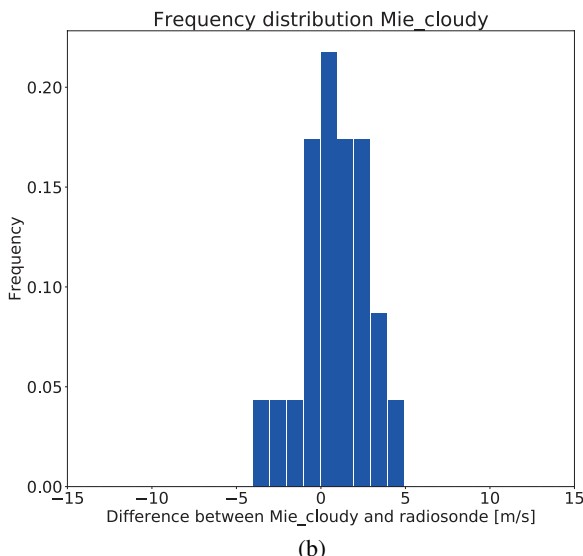

**Figure 11.** a) L2B Mie$_{cloudy}$ winds vs. the radiosonde measurements made during the ship-borne validation except 27 November 2018. b) Frequency distribution of the difference between Mie$_{cloudy}$ and radiosonde wind speeds for the same data set. Radiosonde data is aggregated to the Aeolus vertical resolution and projected to the HLOS of Aeolus.

Figure 10b shows the normalized frequency distribution of the deviation between the Rayleigh$_{clear}$ and radiosonde wind
observations. Considering the relatively small amount of measurements for this statistics, an almost Gaussian shaped distribution is found. Thus, one can conclude that the deviation between Aeolus and the radiosonde wind observation is normally distributed. When calculating the mean value of this distribution, one gets 1.52 m/s as bias for the Rayleigh$_{clear}$ wind observations. If one uses the median of the distribution for the bias calculation, one gets a bias of 1.47 m/s and thus a little less than as calculated from the mean. If one forces the linear regression to have a slope of 1, the retrieved offset is practically the
same as the mean deviation between the radiosonde and Aeolus. As this is expected for a Gaussian distribution, it confirms the normally distributed behaviour of the Rayleigh$_{clear}$ wind deviations.

To conclude, a bias (systematic error) of 1.47–1.57 m/s was derived from the five radiosonde ascents for the Rayleigh$_{clear}$ winds regardless of the calculation method. The median absolute deviation (MAD) of the distribution represents the random error of the Aeolus wind observations (Lux et al., 2020; Witschas et al., 2020). It is in case of the Rayleigh clear winds 3.33 m/s.
The same statistics are shown for the Mie$_{cloudy}$ winds in Fig. 11. Here a much smaller number of points for comparison could be used as Mie$_{cloudy}$ winds are only available at heights where clouds occur. Regardless of the low number of values, it is already obvious that the bias is less than for the Rayleigh$_{clear}$ wind. The bias obtained with the linear regression is relatively high with 1.13 compared to the bias obtained from the median and mean (0.95 and 0.95 m/s, respectively) of the frequency distribution of the differences between Aeolus and the radiosonde (Fig.11 b). Nevertheless, forcing the slope to be unity, the



**Table 2.** Overview of derived biases (systematic errors) and median absolute deviation (MAD - random error) for the Rayleigh$_\text{clear}$ and Mie$_\text{cloudy}$ winds obtained from the comparison with the 5 radiosonde launches. All values in m/s.

| Wind type | mean bias | median bias | regression bias | regression slope | MAD |
|---|---|---|---|---|---|
| Rayleigh$_\text{clear}$ | 1.52 | 1.47 | 1.57 | 0.97 | 3.26 |
| Mie$_\text{cloudy}$ | 0.95 | 0.88 | 1.13 | 0.95 | 1.06 |

same offset (i.e., bias or systematic error) as for the mean of the distribution is retrieved which again confirms that the deviations of Mie$_\text{cloudy}$ winds are normally (Gaussian) distributed.

Figure 11b shows, that the absolute deviations are much lower than for the Rayleigh$_\text{clear}$ winds which is reflected in the much smaller MAD of 1.06 m/s. All absolute deviations are below 5 m/s compared to values up to 13 m/s for the Rayleigh$_\text{clear}$ winds. This is caused by the generally lower Rayleigh return signal compared to the Mie channel. Rayleigh scattering is orders

of magnitude lower than the Mie scattering.

Thus, one can conclude for a user's perspective, that Mie$_\text{cloudy}$ winds are more accurate than Rayleigh$_\text{clear}$ winds (less systematic and less random error) and should be used if available. The Rayleigh$_\text{clear}$ winds, on the other hand, provide a better coverage of the atmosphere.

An overview of the derived values for the systematic and random errors of Aeolus from the ship-borne validation is given

in Tab. 2. The systematic and random errors observed are higher than demanded in the mission requirements (Ingmann and Straume, 2016). A systematic error of less than 0.7 m/s shall be achieved with an overall precision of 1 m/s in the PBL, 2.5 m/s in the troposphere and 3–5 m/s in the stratosphere. However, it is worth to mention again that the Aeolus data which was used is not yet the finalized data set for this space mission. Some instrumental effects, like the hot pixel issue, have not yet been corrected and processor updates with several improvements are expected in the future. Despite the mission requirements could

not yet be achieved, the mission can be seen as success as it was already demonstrated that winds are globally observable from space by active remote sensing with an accuracy needed for assimilation in NWP.

Considering that only five radiosonde launches were used, the observed biases are in agreement with other CAL/VAL teams of this mission (ESA, 2019b): At the first Aeolus CAL/VAL workshop, independent comparisons (not publicly accessible) of several CAL/VAL teams showed global biases in the range of <1 m/s up to 3.3 m/s using different observation periods and

NWP models (e.g. Rennie and Isaksen (2019)).

Lux et al. (2020) compared the Aeolus Rayleigh$_\text{clear}$ wind observations to winds measured with the airborne demonstrator during Windval III in central Europe from 17 November to 5 December 2018 and thus in the same time period as the validation measurements obtained on-board *RV Polarstern*. The authors also validated these winds against wind data from the ECMWF model. They report a bias of 1.6 m/s with random errors of 2.5 m/s (comparison against ECMWF model winds), and 2.53 m/s

bias and 3.57 m/s random error for the comparison against the airborne demonstrator.

Witschas et al. (2020) reported slightly different biases (systematic errors) of 2.1 and 2.3 m/s for Rayleigh$_\text{clear}$ and Mie$_\text{cloudy}$ winds during the same campaign (Wind Val III) using a 2-micrometer airborne Doppler wind lidar. Random errors were of




about 4 m/s and 2.2 m/s for Rayleigh$_{clear}$ and Mie$_{cloudy}$, respectively, in this study. For another campaign, namely AVATARE (Aeolus Validation Through Airborne Lidars in Europe) in central Europe in spring 2019, however, the authors found negative

biases of -4.6 m/s for Rayleigh$_{clear}$ winds and -0.2 m/s for Mie$_{cloudy}$ and an increased random error of 4.4 m/s and 2.2 m/s for Rayleigh$_{clear}$ and Mie$_{cloudy}$, respectively. They conclude that the shift of the bias could be a result of inadequate and constant calibration used during the L2B processing until 16 May 2019 not accounting for instrumental drifts that were observed since launch (Reitebuch et al., 2019a). Meanwhile, the calibration has been updated more regularly and instrument drifts are under investigation to be corrected in future processor updates. For further details, the reader may refer to the stated references.

Khaykin et al. (2020) analyzed one wind profile of Aeolus with the Doppler lidar at Observatoire de Haute-Provence and found a good agreement between the two measurements. But below 5 km a.g.l. (above ground level), a stronger deviation was observed which was considered to be caused by horizontal heterogeneity. In our study, however, we could almost never observe any Rayleigh$_{clear}$ wind profile below 4 km which prohibits the discussion of this issue raised by Khaykin et al. (2020). Nevertheless, it could already be an indicator that the laser energy which has been lower than expected (Kanitz et al., 2019b;

Reitebuch et al., 2019a), leads to less accuracy and therefore more invalid wind observation close to the ground (further away from the lidar on-board Aeolus)

To summarize, the statistics obtained during cruise PS116 with *RV Polarstern*, even if only consisting of five comparisons with radiosondes, do agree well with findings from other CAL/VAL teams and give an insight of the Aeolus performance shortly after launch – thus still in the commissioning phase. It also shows that Aeolus is able to measure horizontal wind

speeds from space and that the retrieved data is reliable within a given uncertainty range and thus ready for data assimilation. First data assimilation experiments have already shown a positive impact, e.g. as announced by ECMWF (ECMWF, 2019a, b).

## 5 Conclusions

Wind products from the first wind lidar in space, ALADIN, on-board the European satellite Aeolus were validated against wind profiles obtained from radiosonde launches on-board the German *RV Polarstern* during the cruise PS116 in Autumn 2018

across the Atlantic Ocean. Six points of intersection were reached within a radius of 150 km for which additional radiosondes could be launched in time. These unique validation measurements across the Atlantic Ocean are a valuable contribution to the – until now – mainly model-based validations of Aeolus in that region of the Earth.

With the analysis of dedicated case studies, it was shown, that Aeolus is able to measure accurately atmospheric wind profiles of the nearly west-east wind component. Due to its HSRL technique, Aeolus is able to measure wind speed in, both,

clear, particle free atmospheric regions and in regions where clouds or dense aerosol layers occur. The corresponding products are the Rayleigh$_{clear}$ and Mie$_{cloudy}$ winds, respectively.

Aeolus, i.e., ALADIN, is able to obtain the shape of the wind profile and the magnitude of the wind speed with sufficient accuracy taking into account also the representativeness error introduced by the horizontal distance between the radiosonde and Aeolus ground track and the low horizontal (87 km) and vertical resolution (0.5–2 km) of Aeolus. A proof of concept of the

HSR Doppler wind lidar technique in space to measure global wind profiles was therefore already demonstrated. Nevertheless,





it was also shown that the height resolution which was set during the commissioning phase was not sufficient to capture the maximum wind speeds in relatively thin strong-wind regions, here discussed in terms of the example of the tropical jet stream. The coarse resolution of Aeolus of 2 km at altitudes above 13 km caused a significant underestimation of the maximum wind speed. Thus, considering the vertical binning, the Aeolus observations were correct, but important information on the tropical

jet stream speed were missing. As a consequence, the range-bins were changed to a resolution of 1 km up to an altitude of 19 km on 26 February 2019 to provide the NWP models a much more detailed wind information in such an important atmospheric region.

It has also been discussed that Rayleigh$_{clear}$ winds in the free troposphere have a larger offset, i.e. systematic error, than the corresponding Mie$_{cloudy}$ winds leading to a slight overestimation of the true HLOS wind speed. Mie$_{cloudy}$ winds are only

available at atmospheric regions with clouds, but the comparison to the radiosonde profiles shows that the Mie$_{cloudy}$ winds were very accurate, with lower systematic and random errors than the Rayleigh$_{clear}$ winds, and should be used when available in favour of the Rayleigh$_{clear}$ winds. Nevertheless, especially the Rayleigh$_{clear}$ winds are a special highlight of the Aeolus mission as they could close a gap for clear air observations in the global atmospheric observing system which are not covered by atmospheric motion vectors obtained in cloudy regions only.

The statistical analysis based on only five radiosondes reveals a good performance of Aeolus in this early phase of the mission having a systematic error (bias) of about 1.5 m/s and random error of 3.3 m/s for the Rayleigh$_{clear}$ winds. The Mie$_{cloudy}$ winds were more accurate with about 1 m/s systematic error and a random error of 1 m/s. This is yet higher than claimed in the mission requirements but it should be noted that the data used for validation here is not the final wind data set of Aeolus. Some known instrumental effects and calibrations have not yet been implemented in the retrieval algorithms. The main challenges of

the Aeolus mission are the occurrence of hot pixel, varying biases, the laser energy development, and the lower atmospheric return signal resulting in a larger Rayleigh random error. ESA is steadily working on the improvements of the wind retrievals and processor updates. Several reprocessing steps of the existing data will take place in the future delivering data with even higher accuracy than the current data set delivered in the commissioning phase of Aeolus.

To summarize, the validation efforts performed with radiosondes launched during cruise PS116 of *RV Polarstern* give an

insight of the Aeolus performance shortly after launch and thus still in the commissioning phase of Aeolus. It shows that Aeolus is able to measure horizontal wind speeds from space and that the retrieved data is reliable within a given uncertainty range and is usable for data assimilation. As announced by ECMWF (ECMWF, 2019b), first data assimilation experiments have already shown a positive impact. For such experiments, the systematic errors obtained during the CAL/VAL efforts are a prerequisite because they need to be corrected and show the importance of independent CAL/VAL activities. Since the beginning of 2020,

Aeolus data is even operationally assimilated at ECMWF (ECMWF, 2020a) and a positive impact on the weather prediction has been shown (Rennie and Isaksen, 2019; Isaksen and Rennie, 2019). The recent global shut down due to the COVID-19 epidemic has even shown that Aeolus is able to partly replace the missing aircraft measurements in the global data assimilation system (ECMWF, 2020b).



*Data availability.* Radiosonde data are available at the PANGAEA Data Center: https://doi.pangaea.de/10.1594/PANGAEA.903888. Aeolus
data used in this publication is not yet freely available but will become public in the near future after re-processing has been performed. Since
May 2020, Aeolus data is publicly available at the ESA Aeolus Online Dissemination System.

*Author contributions.* All authors have contributed to the manuscript preparation. HB and AH have performed the data analysis. AH and KO
performed the measurements on-board RV Polarstern. UW and BH have contribution to the discussion with their expertise in remote sensing
and meteorology. HB has led the manuscript preparation based on the Master thesis of AH.

*Competing interests.* The authors declare no conflict of interest

*Disclaimer.* The presented work includes preliminary data (not fully calibrated/validated and not yet publicly released) of the Aeolus mission
that is part of the European Space Agency (ESA) Earth Explorer Programme. Further data quality improvements, including in particular a
significant product bias reduction, will be achieved before the public data release. The analysis has been performed in the frame of the Aeolus
Scientific Calibration and Validation Team (ACVT).

*Acknowledgements.* This work was funded by the German Federal Ministry for Economic Affairs and Energy (BMWi) under grant no.
(FKZ) 50EE1721C. Authors acknowledge also support through ACTRIS-2 under grant agreement no. 654109 from the European Union's
Horizon 2020 research and innovation programme. We thank the Alfred-Wegener Institute and the RV Polarstern crew for their incredible
effort in making those measurements possible (acknowledgment no. AWI_PS116_00). Furthermore, we thank the German Weather Service
(DWD) for the support in launching radiosondes during the cruise. We also appreciate very much the fruitful discussions within the EVAA
consortium (LMU, DWD, DLR) and with ESA.



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
