# Peer review of "Validation of Aeolus wind products above the Atlantic Ocean"

_Atmospheric Measurement Techniques, 2020_

## Referee Comment (RC1) · Anonymous Referee #1 · 8 Jun 2020

The authors present a unique and highly valuable data set, based on radiosonde launches from the Polarstern cruise across the Atlantic, to assess the performance of Aeolus in the early phase of the mission (autumn 2018). The paper is very clearly written and the results are important to validate the unique Aeolus mission. Still, some corrections are needed upon publication.

Major comments ==============

line 2: Aeolus measures horizontal wind profiles => Aeolus measures profiles of a single horizontal wind component

line 127: 30 x 2.7 = 81 not 87, please correct. It is more like 30 x 2.85.

line 152: "It is obvious that only two out of this four wind products are useful, namely the
Rayleighclear and the Miecloudy product." Well, this is not that obvious. Also Rayleigh-cloudy can provide a useful wind because the L2B can correct for Mie contamination, in principle, using the scattering ratio as input. It is true that until now this has not been succesful enough for operational use. So please update your text accordingly.

Figure 2 is from Raman-polarization lidar Polly, I guess? Please mention in the caption of the figure.

line 133: "In contrast, it was obviously not detected by the Aeolus measurement due to the fact that the Miecloudy wind is obtained practically only from return signals of the cloud and thus only from the height range at which the cloud was observed within this one range bin of 1 km thickness." I guess the problem here is that Aeolus only measures the wind at cloud top, so for a fair comparison you should compare with the radiosonde value at the cloud top. On the other hand, Aeolus cannot determine the exact location of the cloud top inside the vertical bin and the best one can do is to assign the Mie wind to the bin centre location, hence giving the large error the authors observe. Note that the Mie channel is much more sensitive to such height assignment errors than the Rayleigh channel (X. J. Sun, R. W. Zhang, G. J. Marseille, A. Stoffelen, D. Donovan, L. Liu, and J. Zhao, The performance of Aeolus in heterogeneous atmo-spheric conditions using high-resolution radiosonde data, Atmos. Meas. Tech., 7 , pp. 2695-2717, 2014, doi:10.5194/amt-7-2695-2014)

line 268: "Thus, these wind measurements at this altitude should be neglected until the hot pixel correction is in place." Since figure 5b does not show altitude, it would be good to explicitly mention to ignore the (dark) red colors, which indeed contradict with the Rayleigh-clear winds above (discontinuity).

line 318: "Due to its large vertical resolution". This is incorrect. Should be "coarse vertical resolution" or "large vertical bins"

Figure10a/11a. Data analysis from Aeolus have shown substantial differences between statistics from ascending and descending orbits. In figure 10a and 11a it would be

interesting to indicate this by using different colors for the dots in the scatterplot.

line 345: "Considering the relatively small amount of measurements for this statistics, an almost Gaussian shaped distribution is found. Thus, one can conclude that the deviation between Aeolus and the radiosonde wind observation is normally distributed." You cannot conclude this based on the shape of the distribution in figure 10b. Please rephrase or otherwise apply a statistical analysis to test this Gaussian hypothesis.

line 353: Please mention that the MAD is less sensitive to outliers and equals 0.674 x STD for a perfectly Gaussian distributed stochastic variable. This gives a good handle on how to interpret the value of 3.33 m/s in the next line, i.e., it corresponds to 4.94 m/s error standard deviation, the metric more commonly used in error quantification and data assimilation.

Table 2. Why differs the value of 3.26 for Rayleigh-clear MAD from 3.33 in the text? Please correct.

line 364: "This is caused by the generally lower Rayleigh return signal compared to the Mie channel. Rayleigh scattering is orders of magnitude lower than the Mie scattering." The difference is not only SNR, it is also different interferometers and different type of processing for wind retrieval (peak fitting versus fitting of measured Rayleigh Response to temperature and pressure dependent (Rayleigh Response, Doppler shift) curves/tables). Please rephrase to something like: "This is mainly caused by the generally lower SNR of the Rayleigh return signal compared to the Mie channel, besides the different measurement and retrieval techniques".

line 373: "Some instrumental effects, like the hot pixel issue, have not yet been corrected" In the mean time, the main reason for biases of Rayleigh winds has been found: temperature variations over the telescope which are not fully compensated for by the instrument. Please mention this too.

line 374: "Despite the mission requirements could not yet be achieved, the mission can

be seen as success as it was already demonstrated that winds are globally observable from space by active remote sensing with an accuracy needed for assimilation in NWP." I would end with "........ active remote sensing with sufficient quality to demonstrate positive impact in NWP [Ref]". With a reference to results presented by ECMWF or ESA outreach publications.

line 397: "horizontal heterogeneity". I guess you mean: "horizontal atmospheric heterogeneity"? Please add.

line 421: "was not sufficient to capture the maximum wind speeds in relatively thin strong-wind regions, here discussed in terms of the example of the tropical jet stream". I would rephrase to: "was not sufficient to capture events of strong vertical wind-shear such as near the tropical jet stream".

line 434: you could add: "in fact, Rayleigh-clear winds have proven more beneficial for NWP than Mie-cloudy winds".

line 436: "..... and random error of 3.3 m/s for the Rayleigh ...." This is very misleading as this value does not represent the usual STD but MAD, see comment above. So please translate this value to STD.

line 438: "Some known instrumental effects and calibrations have not yet been implemented in the retrieval algorithms" Rephrase to: "In the mean time discovered instrumental and calibration imperfections were not yet implemented in the retrieval algorithms used for the 2018 autumn data set"

In this context, do you have plans to use reprocessed, unbiased, Aeolus data with the same radiosonde data set presented here? I would very much encourage the authors to write a follow-up paper, once the reprocessed data for the autumn 2018 period become available. If so, please mention in section 5.

Minor comments / typos =======================

line 39: observation => observations line 62: chosen => selected line 89: around ≈;

remove either 'around' or '≈' line 94: to retrieve wind retrievals => to retrieve winds line 100: correction => corrections line 110: the data must be available within 3 hours => the data must be available within 3 hours after measurement time (timeliness). By the way, this is not true for ECMWF who wait about 5 hours before they start there analysis run. This is valid for medium-range forecasts. Mesoscale meteo centers need the data within 3 hours for operational use. line 116: parameter => parameters line 129: The currently applied method by ESA is the use of the scattering ratio => The currently applied method by ESA is the use of the scattering ratio, which is determined as part of the L1B processing (ref) and used as input for the L2B processing. line 135: comprised => comprises line 159: please explain DISC line 164/166: pixel => pixels line 166: increase => increased line 246: the resolution is simply too low => the resolution is simply too coarse
* * *

---

## Referee Comment (RC2) · Anonymous Referee #2 · 22 Jul 2020

This is a good paper that makes an important contribution by investigating direct comparisons of Aeolus wind measurements with radiosonde-observed winds over the ocean. Although the number of cases is limited, nevertheless the conclusions from the comparisons are self-consistent and demonstrate the important conclusion that Aeolus is making acceptable and representative measurements of the horizontal line of sight winds. The point illustrated from the analysis that the large vertical range bins in the upper troposphere lead to underestimate of the maximum wind in regions where the vertical shear is high is important. I believe that the paper is suitable for publication following correction and clarification of a few minor issues that I describe below.

Line 10: "proof" should be "prove" or "provide proof".

Line 34: It should be noted that Atmospheric Motion Vectors (AMVs) can also be com-

puted by tracking features in the water vapor field.

Line 153: It isn't obvious to me that the Rayleighcloudy wind product is without value. I realize that the presence of aerosols complicates the wind retrieval in the Rayleigh channel, but I'm not aware that the Rayleighcloudy product is deemed totally useless. If the Aeolus project team has stated this then the authors should provide a reference.

Lines 157-162: Although references are provided, as a reader it would be nice to have a few sentences describing in general terms how the error threshold and validity flags are computed.

Line 200: Perhaps I missed it, but it would be useful to state in the text that because the Aeolus lidar beam is not nadir-pointing, the horizontal distance from the radiosonde to the Aeolus measurement volumes changes as a function of height as well as radiosonde movement. It's a simple and obvious point, but it can't hurt to note it.

Figure 3 and Figure 7: In looking at the figures on a laptop, I found it somewhat difficult to differentiate the colors in the Mie cloud and Rayleigh clear plots. Perhaps the authors could use a different technique for separating the plots, such as dashed or dotted lines.

Line 247: The inability of Aeolus to characterize the maximum wind under strong shear conditions near the tropopause is useful to point out. However, it should probably be noted that this isn't an error in the Aeolus measurement, but rather an averaging effect that obscures an important parameter.

Line 253: Changing the range bins on Aeolus to 1 km has potentially negative consequences on the measurement in that it reduces the number of photons available, thus increasing the random error. The authors might want to comment on whether the Aeolus team chose to accept this increase in random error or compensate for it by, e.g., reducing the horizontal resolution.

Figure 5: it would be nice to provide a N-S reference on the plots.

Line 353: A sentence explaining why the authors prefer to use MAD as the statistic to

represent the random error would be useful.

Table 2 Caption: The caption seems to be defining medium absolute deviation (MAD) as MAD - random error, which doesn't make sense.

---

## Author Comment (AC1) · 31 Jul 2020

We thank reviewer 1 very much for his/her careful reading and the corresponding valuable comments and suggestions, which are for sure very beneficial for the presented paper. We agreed on almost all. Please find below our reply to each comment (*italic*). Major comments if Reviewer 1

- line 2: Aeolus measures horizontal wind profiles => Aeolus measures profiles of a single horizontal wind component
  *We changed the text accordingly.*

- line 127: 30 x 2.7 = 81 not 87, please correct. It is more like 30 x 2.85.
  *Thanks for the careful reading. We changed it accordingly and gave reference to the recent memorandum for wind retrievals (doi:10.21957/alift7mhr)*

- line 152: "It is obvious that only two out of this four wind products are useful, namely the Rayleighclear and the Miecloudy product." Well, this is not that obvious. Also Rayleigh- cloudy can provide a useful wind because the L2B can correct for Mie contamination, in principle, using the scattering ratio as input. It is true that until now this has not been successful enough for operational use. So please update your text accordingly.

  *You are completely right. I even have analysed the Rayleigh_cloudy winds by myself and found partly good agreement. Thus, text was changed accordingly.*

- Figure 2 is from Raman-polarization lidar Polly, I guess? Please mention in the caption of the figure.
  *Done!*

- line **2**33: "In contrast, it was obviously not detected by the Aeolus measurement due to the fact that the Miecloudy wind is obtained practically only from return signals of the cloud and thus only from the height range at which the cloud was observed within this one range bin of 1 km thickness." I guess the problem here is that Aeolus only measures the wind at cloud top, so for a fair comparison you should compare with the radiosonde value at the cloud top. On the other hand, Aeolus cannot determine the exact location of the cloud top inside the vertical bin and the best one can do is to assign the Mie wind to the bin centre location, hence giving the large error the authors observe. Note that the Mie channel is much more sensitive to such height assignment errors than the Rayleigh channel (X. J. Sun, R. W. Zhang, G. J. Marseille, A. Stoffelen, D. Donovan, L. Liu, and J. Zhao, The performance of Aeolus in heterogeneous atmo spheric conditions using high-resolution radiosonde data, Atmos. Meas. Tech., 7 , pp. 2695-2717, 2014, doi:10.5194/amt-7-2695-2014)
  *You are right with that. We meant the same, but obviously our non-native English prohibited the right message. We change this accordingly:*
  *"The disagreement is caused because with the Aeolus Mie algorithm, the wind speed at cloud top is measured but due to the range-bin thickness of 1 km, the top height of this cloud cannot be correctly assigned. Thus, the Mie wind speed measured at cloud top is assigned to the center of the 1 km thick range-bin disregarding the true top-height of the cloud. As a consequence, the agreement to the high resolution radio sonde profile is much better (almost identical values at 2.5 km) than to the radio sonde profiles binned to Aeolus resolution. The presence of cloud or aerosol layers in the measurement bins was already discussed prior launch by Sun et al. (2014) and it was shown that biases of more than 0.4 m/s can occur when the cloud top is not in the center of the range-bin. This statement is confirmed by our observations and shows that a higher vertical resolution is in principle preferable and valuable."*

- line 268: "Thus, these wind measurements at this altitude should be neglected until the hot pixel correction is in place." Since figure 5b does not show altitude, it would be good to explicitly

mention to ignore the (dark) red colors, which indeed contradict with the Rayleigh-clear winds above (discontinuity).
*We added: "(indicated by reddish colors just above the bluish colors in the lowermost profile)".*

- line 318: "Due to its large vertical resolution". This is incorrect. Should be "coarse vertical resolution" or "large vertical bins"
*corrected!*

  - Figure10a/11a. Data analysis from Aeolus have shown substantial differences between statistics from ascending and descending orbits. In figure 10a and 11a it would be interesting to indicate this by using different colors for the dots in the scatterplot.
  *Thanks for this very interesting advice. We updated the Figures accordingly and also added one more column in Table 1 indicating either ascending or descending orbit. Nevertheless, from our 6 observations in this early mission stage, no significant difference between the two orbit types are visible.*
  *We added in the text: "The different colors indicate whether Aeolus had an ascending node (green) or descending node (red), i.e., if the measurement was taken at local evening or local morning, respectively. This separation is done because first long-term Cal/Val activities showed significantly differences in the determined biases of Aeolus wind measurements between the two different modes (Rennie and Isaksen, 2020; Geiß et al., 2019; Krisch et al., 2020). However, from our observations onboard RV Polarstern in the early mission phase of Aeolus, we do not observe a significant difference between the two modes with respect to the Rayleigh_clear winds." and regarding the Mie winds:*
  *"As for the Rayleigh_clear winds, no difference between the Aeolus performance for ascending and descending orbits is found (Fig.11a)."*

- line 345: "Considering the relatively small amount of measurements for this statistic, an almost Gaussian shaped distribution is found. Thus, one can conclude that the deviation between Aeolus and the radiosonde wind observation is normally distributed." You cannot conclude this based on the shape of the distribution in figure 10b. Please rephrase or otherwise apply a statistical analysis to test this Gaussian hypothesis.

  *You are right, we deleted this statement at this stage and modified a later statement about this, so that the conclusion is significantly weakened: "As this is expected for a Gaussian distribution, one could assume, in accordance with the shape of the distribution shown in Fig. 10b, a normally distributed behaviour of the Rayleigh_clear wind deviations."*

- line 353: Please mention that the MAD is less sensitive to outliers and equals 0.674 x STD for a perfectly Gaussian distributed stochastic variable. This gives a good handle on how to interpret the value of 3.33 m/s in the next line, i.e., it corresponds to 4.94 m/s error standard deviation, the metric more commonly used in error quantification and data assimilation.
*Thanks, done:*
*"The median absolute deviation (MAD) of the distribution is used to calculate the random error of the Aeolus wind observations (Lux et al., 2020; Witschas et al., 2020) because it is less sensitive to outliers than the standard deviation. It is 67.4% of the standard deviation or the other way around, the scaled MAD (MAD times 1.4826) is identical to the standard deviation for a perfectly Gaussian distribution. The scaled MAD is thus used an indicator for the random error for Aeolus observations. The MAD is in case of the Rayleigh clear winds 3.26 m/s, the scaled MAD correspondingly 4.84 m/s."*

- Table 2. Why differs the value of 3.26 for Rayleigh-clear MAD from 3.33 in the text? Please correct.
  *Done, thanks*

- line 364: "This is caused by the generally lower Rayleigh return signal compared to the Mie channel. Rayleigh scattering is orders of magnitude lower than the Mie scattering." The difference is not only SNR, it is also different interferometers and different type of processing for wind retrieval (peak fitting versus fitting of measured Rayleigh Response to temperature and pressure dependent (Rayleigh Response, Doppler shift) curves/tables). Please rephrase to something like: "This is mainly caused by the generally lower SNR of the Rayleigh return signal compared to the Mie channel, besides the different measurement and retrieval techniques".

  *Done! Thanks for the suggestion!*

- line 373: "Some instrumental effects, like the hot pixel issue, have not yet been corrected" In the mean time, the main reason for biases of Rayleigh winds has been found: temperature variations over the telescope which are not fully compensated for by the instrument. Please mention this tool
  *We mentioned it and rephrased to:*
  *"Despite the mission requirements could not yet be achieved, the mission can be seen as success as it was already demonstrated that winds are globally observable from space by active remote sensing with sufficient quality to achieve a positive impact in NWP (Rennie and Isaksen, 2020; Martin et al., 2020). However, it is worth to mention again that the Aeolus data which was used is not yet the finalized data set for this space mission. In the meanwhile it was found, that slight temperature variations over the receiving telescope area are one of the main reasons for biases of the Rayleigh winds (Rennie and Isaksen, 2020; Krisch et al., 2020; Reitebuch et al., 2020). This effect and some other instrumental challenges, like the hot pixels issue, have not yet been compensated in the data of the early mission stage. Processor updates with several improvements have been taking place in the meantime and more are expected in the future to correct such effects, after which a reprocessing of the early Aeolus data set is foreseen.*

- line 374: "Despite the mission requirements could not yet be achieved, the mission can be seen as success as it was already demonstrated that winds are globally observable from space by active remote sensing with an accuracy needed for assimilation in NWP." I would end with "........ active remote sensing with sufficient quality to demonstrate positive impact in NWP [Ref]". With a reference to results presented by ECMWF or ESA outreach publications.

  *DONE, see above.*

- line 397: "horizontal heterogeneity". I guess you mean: "horizontal atmospheric heterogeneity"? Please add.
  *Thanks!*

- line 421:"was not sufficient to capture the maximum wind speeds in relatively thin strong-wind regions, here discussed in terms of the example of the tropical jet stream". I would rephrase to: "was not sufficient to capture events of strong vertical wind-shear such as near the tropical jet stream". *Thanks for the suggestion! Sone!*

- line 434: you could add: "in fact, Rayleigh-clear winds have proven more beneficial for NWP than Mie-cloudy winds".
  *Added. Thanks!*

- line 436: "..... and random error of 3.3 m/s for the Rayleigh ...." This is very misleading as this value does not represent the usual STD but MAD, see comment above. So please translate this value to STD.
  *Thanks for the hint! Done!*

- line 438: "Some known instrumental effects and calibrations have not yet been implemented in the retrieval algorithms" Rephrase to: "In the meantime discovered instrumental and calibration imperfections were not yet implemented in the retrieval algorithms used for the 2018 autumn data set"
  *Done!*

- In this context, do you have plans to use reprocessed, unbiased, Aeolus data with the same radiosonde data set presented here? I would very much encourage the authors to write a follow-up paper, once the reprocessed data for the autumn 2018 period become available. If so, please mention in section 5.
  *Yes, this is a good idea. We mentioned it now: "Once a final reprocessing has been taken place it could be worth to use the existent RV Polarstern data set to quantify the improvements of the algorithm updates."*

Minor comments / typos ======================

- line 39: observation => observations*, done*

- line 62: chosen => selected *, done*

- line 89: around ≈;remove either 'around' or '≈' , done

- line 94: to retrieve wind retrievals => to retrieve winds*, done*

- line 100: correction => corrections*, done*

- line 110: the data must be available within 3 hours=> the data must be available within 3 hours after measurement time (timeliness). By the way, this is not true for ECMWF who wait about 5 hours before they start there analysis run. This is valid for medium-range forecasts. Mesoscale meteo centers need the data within 3 hours for operational use. , *thanks, good to know!*

- line 116: parameter => parameters *, done*

- line129: The currently applied method by ESA is the use of the scattering ratio => The currently applied method by ESA is the use of the scattering ratio, which is determined as part of the L1B processing (ref) and used as input for the L2B processing. *, done*

- Line135: comprised => comprises *, done*

- line 159: please explain DISC *, done*

- line 164/166: pixel => pixels , *done!*

- line 166: increase => increased *, done*

- line 246: the resolution is simply too low => the resolution is simply too coarse, *done*

---

## Author Comment (AC2) · 31 Jul 2020

We thank reviewers 2 for his/her time and the valuable comments and suggestions. Please find below the response from us indicated in *italic*.

- Line 10: "proof" should be "prove" or "provide proof".
  *Thanks – changed!*

- Line 34: It should be noted that Atmospheric Motion Vectors (AMVs) can also be computed by tracking features in the water vapor field. *Thanks for this information, we added it to the text and added a new reference (Bormann, N., S. Saarinen, G. Kelly, and J. Thépaut, 2003: The Spatial Structure of Observation Errors in Atmospheric Motion Vectors from Geostationary Satellite Data. Mon. Wea. Rev., 131, 706–718,)*

- Line 153: It isn't obvious to me that the Rayleigh_cloudy wind product is without value. I realize that the presence of aerosols complicates the wind retrieval in the Rayleigh channel, but I'm not aware that the Rayleigh_cloudy product is deemed totally useless. If the Aeolus project team has stated this then the authors should provide a reference.
  *You are right as reviewer 1 has also mentioned and I have also used this wind type for investigations. We changed the text accordingly:*
  *"Two out of this four wind products, namely the Rayleigh_clear and the Mie_cloudy winds, are the main target for the operational use of Aeolus data in NWP"… "The Rayleigh_cloudy products may also deliver usable wind measurements, but contamination of Mie scattering need to be corrected first which is yet at an experimental stage. Thus, we will use only Rayleighclear and the Miecloudy product for our analysis"*

- Lines 157-162: Although references are provided, as a reader it would be nice to have a few sentences describing in general terms how the error threshold and validity flags are computed.
  *We added some few sentences concerning that. But we think, the full explanation of the validity flag is not needed when the reference is given:*
  *"…These thresholds are chosen subjectively, based on the compromise between the number of observations that pass the quality control and the overall quality of the dataset (Rennie and Isaksen, 2020)." …"The validity flag (de Kloe et al., 2016) considers the validity of the products. Several different technical, instrumental and retrieving checks account for this flag. ."*

- Line 200: Perhaps I missed it, but it would be useful to state in the text that because the Aeolus lidar beam is not nadir-pointing, the horizontal distance from the radiosonde to the Aeolus measurement volumes changes as a function of height as well as radiosonde movement. It's a simple and obvious point, but it can't hurt to note it.
  *We have added this information accordingly: "As Aeolus is not pointing nadir but is taking measurements 35° off-nadir, the horizontal distance of the Aeolus observations to RV Polarstern is different for the different heights in the Aeolus wind profile. Also the radiosonde drifts along the wind direction, thus the distance to between the Aeolus measurements and the radiosonde changes during the ascent. The effect of both is illustrated…"*

- Figure 3 and Figure 7: In looking at the figures on a laptop, I found it somewhat difficult to differentiate the colors in the Mie cloud and Rayleigh clear plots. Perhaps the authors could use a different technique for separating the plots, such as dashed or dotted lines.
  *Thanks for this advise. We reshaped all corresponding figures accordingly so that colors are not needed anymore and we hope that they are now more clearly readable.*

- Line 247: The inability of Aeolus to characterize the maximum wind under strong shear conditions near the tropopause is useful to point out. However, it should probably be noted that this isn't an error in the Aeolus measurement, but rather an averaging effect that obscures an important

parameter.
*You are right, we've added: "This is in principle no measurement error of Aeolus."*

- Line 253: Changing the range bins on Aeolus to 1 km has potentially negative consequences on the measurement in that it reduces the number of photons available, thus increasing the random error. The authors might want to comment on whether the Aeolus team chose to accept this increase in random error or compensate for it by, e.g., reducing the horizontal resolution.
*This is a good point. But as we are "only" a Cal/Val team and not any decision-making body, we would not like to comment too much on these issues. Nevertheless, we've added: "…but accepting the drawback of an increased random error."*

- Figure 5: it would be nice to provide a N-S reference on the plots.
*Thanks, we have added this to the plots!*

- Line 353: A sentence explaining why the authors prefer to use MAD as the statistic to represent the random error would be useful.
*As also raised by reviewer 1, we meanwhile provide the scaled MAD as an indicator for the random error. This is explained in the text and also that the MAD and thus also the scaled MAD is less sensitive to outliers in contrast to the standard deviation:*
*"The median absolute deviation (MAD) of the distribution is used to calculate the random error of the Aeolus wind observations (Lux et al., 2020; Witschas et al., 2020) because it is less sensitive to outliers than the standard deviation. It is 67.4% of the standard deviation or the other way around, the scaled MAD (MAD times 1.4826) is identical to the standard deviation for a perfectly Gaussian distribution. The scaled MAD is thus used an indicator for the random error for Aeolus observations. The MAD is in case of the Rayleigh clear winds 3.26 m/s, the scaled MAD correspondingly 4.84 m/s.*

- Table 2 Caption: The caption seems to be defining medium absolute deviation (MAD) as MAD - random error, which doesn't make sense.
*You are right. Reviewer 1 also raised this point. Thus, we have added a column for the scaled MAD which is representative for the random error and rephrased the caption accordingly.*